# Foster children's perspectives on participation in child welfare processes: A meta-synthesis of qualitative studies

Jill R. McTavish[ID]¹ᵒ*, Christine McKee¹ᵒ, Harriet L. MacMillan¹,²ᵒ

1 Department of Psychiatry and Behavioural Neurosciences, McMaster University, Hamilton, ON, Canada,
2 Department of Pediatrics, McMaster University, Hamilton, ON, Canada

ᵒ These authors contributed equally to this work.
* mctavisj@mcmaster.ca

## Abstract

The objective of this meta-synthesis was to systematically synthesise qualitative research that explores foster children's perspectives on participation in child welfare processes. Searches were conducted in Medline (OVID), Embase, PsycINFO, and Social Science Citation Index. Children in non-kinship foster care in any setting (high-income, middle-income, low-income countries) who self-reported their experiences of care (removal from home, foster family processes, placement breakdown) were eligible for inclusion. Selected studies took place in 11 high-income countries. A total of 8436 citations were identified and 25 articles were included in this meta-synthesis. Studies summarized the views of 376 children. Children had been in foster care between two weeks and 17 years. Findings synthesize 'facets' of children's participation (e.g., being asked vs making decisions), as well as children's perceived barriers and facilitators to participation. A main priority for children was the quality of their relationships, especially in terms of values (e.g., fairness, honesty, inclusivity). No one way of participating in child welfare processes is better than another, as some children more clearly expressed a desire for passive listening roles and others indicated a desire for active roles in decision-making. However, meaningful adults in foster children's lives have a responsibility to act in a way that strengthens the emphasis on children's needs and voices.

## Introduction

Child maltreatment is a common experience associated with serious adverse outcomes across the lifespan, such as injuries, developmental delay, anxiety and mood disorder symptoms, poor peer relationships, substance use and other risky behaviours [1–5]. A small minority of children who experience maltreatment are removed from their family-of-origin and placed in out-of-home care, including foster care (non-kinship), kinship care, or institutional care [6–9]. It is challenging to assess the benefits and harms of out-of-home care as an intervention for many reasons, in particular, whether the benefits or harms result from differences in a broad range of baseline factors, including socioeconomic status, caregiver educational status,

**Data Availability Statement:** All data relevant to the study are included in the article, uploaded as online supplementary information, or available in Dryad data repository (doi: https://doi.org/10.5061/

dryad.8pk0p2nqs). (Please note: during review process the data is available here: https://datadryad.org/stash/share/DSWnUl_i6XiOAmsQmehksryRzaq3JI7su69eECjBbzs.)

**Funding:** HLM is supported by the Chedoke Health Chair in Child Psychiatry at McMaster University in Hamilton, Ontario, Canada. The funder had no role in study design, data collection and analysis, decision to publish, or preparation of the manuscript.

**Competing interests:** The authors have declared that no competing interests exist.

immigration status, family risks, child welfare worker propensity to place children, and children's safety and well-being at the time of placement [10–15].

In the context of understanding the effects of out-of-home care, there is increasing recognition of the critical importance of assessing children's perspectives on all aspects of out-of-home care [16–19]. Children's right to participate in matters affecting them was established in the Articles 12 and 13 of the United Nations Convention on the Rights of the Child, ratified in 1989. In this paper we summarize the perspectives of children in foster care, with emphasis on their perspectives about participation in child welfare processes (or lack thereof). Child welfare processes generally refer to a set of government and private services primarily designed to protect children from child maltreatment, encourage family stability, and, when necessary, arrange foster care and adoptions. Child protection services is a narrower set of services within child welfare that investigates allegations of child maltreatment. Child welfare processes and level of service response vary by country [20]. For example, most countries allow for voluntary reporting of child maltreatment (with considerable variability in mandatory reporting requirements), require reports to be investigated within a set time period, and also require that some sort of services are provided, such as services for parents (e.g., substance use treatment), for children (e.g., therapy programs), or general services (e.g., universal free medical care) [20]. We focus exclusively on children in non-kinship foster care (hereafter referred to as foster care) given the differences among types of care (e.g., foster care versus kinship care) [21] and our goal of prioritizing the words of children who experience this type of care.

Children's participation often refers to the action of taking part in an activity or decision-making. Sinclair [19] has noted that in practice, children's participation has generally referred to being listened to or consulted. This follows closely from the United Nations Convention on the Rights of the Child articles on participation which emphasize the right to express "views" and "be heard".

An often-cited difficulty of children's participation related to foster care and child protection services processes in general is the tension between safety and participation. There is an ambiguity in policy and practice as to whether children are active "beings" with the right to participation or vulnerable "becomings" in need of protection [22, 23]. Indeed, this tension is reflected in the United Nations Convention on the Rights of the Child itself, as Article 3 acknowledges that children's well-being is reliant on the protection and care of adults. In this meta-synthesis, we summarize children's perspectives on their participation in foster care processes and in particular their (non-) participation in the removal from home, foster family processes, and placement breakdown. The findings of this meta-synthesis will be useful to policy makers, who are increasingly requested to incorporate children's voices into decision-making [16, 24, 25], and practitioners, who need to consider the nuances of when and how to include children's voices in practice and decision-making [26]. The inclusion of children's voices has a number of benefits to children, practitioners, and policy makers, such as affording children their inherent rights to participate, empowering children and reducing their confusion regarding services, and improving service delivery through more tailored, responsive services [24, 26, 27]. As the authors of one review note, children's "participation has both intrinsic (dignity and self-worth in terms of expressing views to influence decisions about their lives) and instrumental (policy and better outcomes for children in terms of supportive relationships with their workers and positive experiences at school and in CPS [children protection services]) value" [24]. This meta-synthesis adds to an important increasing focus in the literature on children's participation across all aspects of child welfare processes [24, 26–28], but with an explicit focus on how foster children speak about their participation.

## Methods

In this paper we followed the methods by Feder et al. [29], whose work built upon Noblit and Hare's [30] approach to meta-ethnography. Specifically, we 1) conducted a systematic search, 2) quality appraised included articles, and 3) inductively analyzed included studies. In line with current critiques about the use of quality appraisal to exclude research with potentially relevant findings [31], we have not excluded any studies based on the results of the critical appraisal. Each one of these methodological choices is discussed further below. The results of this meta-synthesis have been reported according to the PRISMA checklist and Enhancing Transparency in Reporting the Synthesis of Qualitative (ENTREQ) research statement [32] (see S1 Table). A protocol does not exist for this review.

### Search strategy

The systematic search was conducted by an information professional (JRM). Index terms and keywords related to foster care (e.g., foster care, out-of-home care, child protection investigation) and qualitative research (e.g., qualitative, hermeneutics, focus group) were used in the following databases: Medline (OVID), Embase, PsycINFO, Social Science Citation Index (see S1 File example search strategy). Databases were searched for results from the past 20 years, 2000 to November 7, 2019 when the search was executed. The search was updated to February 2, 2021 before submission for publication review. Forward and backward citation chaining was conducted to complement the search. All articles identified by our database searches were screened by two independent reviewers (JRM and CM or HLM) at the title and abstract and full-text level. At the level of title and abstract screening, an article suggested for inclusion by one screener was sufficient to put it forward to full-text review. At the level of full-text, articles with discrepancies were resolved by consensus.

### Study selection criteria

The studies included in this review were a sub-set of a larger review about children's self-reported experiences of foster care. The inclusion criteria for the larger review are as follows: (1) English-language, (2) primary studies that used a qualitative design; (3) published articles; (4) investigations of children's self-reported experiences of foster care, which could include: removal from home, foster family processes, or placement breakdown (see Table 1 for full inclusion criteria of the broader review). It could also include children's self-reported experiences of formal and informal relationships while in foster care, including with social workers, foster carers, peers, or biological family members (siblings, parents, grandparents, aunts/uncles, cousins). The broader review excluded some themes that have been addressed by recent reviews, such as children's self-reported experiences with violence itself (e.g., barriers to disclosure [33]), transitions out of foster care [34], and experiences at school [35]. Within this broader set of included articles, we found considerable discussion of themes related to participation in child welfare processes—specifically, of children's requests to 1) have foster care processes explained to them, 2) be able to speak and to be listened to, 3) be included in decision-making processes, and 4) be able to make their own decisions while in foster care. In order to give space to these important themes, this meta-synthesis presents children's perspectives on participation in child welfare processes.

### Critical appraisal

For critical appraisal, a modified appraisal tool from the Critical Appraisal Skills Programme (CASP) was used to assess the quality of each article [36]. An example of this form is provided

**Table 1. Inclusion and exclusion criteria.**

**Inclusion criteria**

**1. Population:** Children (0<18) who have experienced foster care. At least 80% of the sample must be foster children.

**2. Situation:** Children's self-reported experiences of foster care, which could include: removal from home, foster family processes, or placement breakdown. It could also include children's self-reported experiences of formal and informal relationships while in foster care, including with social workers, foster carers, or biological family members (siblings, parents, grandparents, aunts/uncles, cousins). The study's primary purpose must be to examine children's self-reported reflections on benefits or limitations of these foster care experiences.

**3. Publication type:** Primary, published articles

**4. Study design:** Qualitative study designs that seek to capture children's voices, such as interviews or focus groups. Qualitative study designs ideally included direct quotes from children, but at the very least distinguished children's perspectives in author's summaries. Mixed methods papers were included if qualitative themes from foster children were distinct (e.g., had their own section, were clearly labelled as deriving from qualitative methods).

**5. Setting:** Any setting in low-, middle- and high-income countries

**6. Languages:** English

**7. Timing:** Last 20 years (2000–2020) (Search was updated to 2021 before submission for review.)

**Exclusion criteria**

**1. Ineligible population:** Non-children's perspectives, such as adults; non-foster care experiences, such as experiences in kinship care, residential care, group homes, psychiatric care; non-foster children's perspectives, such as children experiencing depression; children's perspectives on forms of violence, such as bullying, corporal punishment, community violence, or sex trafficking (unless related to their foster care experiences); or accounts of children's perspectives voiced through adults (e.g., forensic interviews). Studies were excluded if they addressed specific programs (e.g., youth advisory boards, specific therapeutic modalities, mentorship) or specific sub-populations (e.g., refugees, runaway youth, cross-over youth).

**2. Ineligible situations:** Articles that focused on foster children's self-reported experiences with violence itself (e.g., barriers to disclosure), with transitions out of foster care, or with their experiences at school or with peers were excluded. Articles that focused on foster children's discussions about their well-being (e.g., perspectives on help-seeking, mental health, sexual health, pregnancy) or about providers supporting their well-being (e.g., mental health providers) were also excluded. Foster children's perspectives on these situations were included if the article primarily focused on one of the situations listed in the inclusion criteria (e.g., if children discussed feelings about removal from home).

**3. Ineligible publication type:** Books, book chapters, reports, dissertations, secondary research (e.g., reviews)

**4. Study design:** Quantitative study designs and qualitative study designs that did not distinguish children's perspectives (e.g., interviewed children and foster carers and did not delineate which themes came from which sample).

**5. Languages:** Non-English

in S2 Table. This modified CASP tool rearranged the questions listed in the original CASP Appraisal Checklist according to standard conceptions of rigour in qualitative research: credibility, transferability, consistency, and neutrality. It also included additional strategies for establishing credibility, transferability, and neutrality that are not discussed in the CASP tool but are found in other discussions of qualitative rigour. One author (JRM) appraised all articles and a second author (CM or HLM) checked each appraisal, with differences in ratings resolved by consensus. Studies were not excluded for poor study design, as we felt that the exclusion of any articles could have excluded a valuable quote/perspective from a child and that this exclusion could impact the meta-synthesis findings.

## Data analysis

Data coding for this meta-synthesis was primarily inductive. Our coding strategy aligns with thematic analysis [37] with an emphasis on the exceptions within children's themes about participation [38, 39]. Thematic analysis involves familiarizing oneself with the data, generating initial codes, searching for themes, reviewing themes, defining and naming themes, and writing up the themes in a manuscript form. All authors involved in coding (CM, JRM, HLM)

initially familiarized themselves with the data through close reading of all the papers and memoing their reflections and reactions to the themes. Authors (CM, JRM, and HLM) then independently placed the primary data from each study and its corresponding code into an Excel file, and these files were compared for consistency (JRM). Primary data, usually found in the Results section, included children's quotes and study authors' summaries of children's words when direct quotes were not available. The first author generated an initial set of codes by descriptively coding the data [37] according to prevailing themes in the child maltreatment literature (e.g., loss, attachment). However, upon discussions with the senior author, we decided to re-code the data focusing more closely on *children's* voices and interpretations of foster care.

Each theme was listed in an evolving word document where themes were collapsed, expanded upon, or abandoned as an iterative process to capture recurring themes related to children's participation. Then, within each theme we examined conditions that maximized or minimized children's participation, somewhat akin to coding conditions for participation [37].

After reviewing discrepancies in themes across Excel files, one author (JRM) developed a master list of codes, and after discussion with other reviewers (CM, HLM) (where all three authors reviewed all codes and corresponding data together), this list of codes was further modified. Any discrepancies identified across the three authors were resolved by consensus. When this final list of themes was developed, which captured commonly recurring themes and exceptions, themes were defined and named. After the development of this master list of codes, one author (JRM) recoded all data in the Excel file according to the revised master list of codes. The final Excel file, which includes all extracted data and codes can be accessed via the Dryad data repository (doi: https://doi.org/10.5061/dryad.8pk0p2nqs).

The authors used several strategies to assess reflexivity during the coding process, such as memoing our reactions to articles (e.g., our reactions to children's voices), discussing differences in perspectives across the authors, and continually reorienting ourselves to children's perspectives on participation. Discussions across the authors served to balance the tone of the findings, or to acknowledge both the potential benefits and limitations of foster care from the perspective of children. Considerable efforts were taken to highlight and prioritize children's expressed understanding of their experiences in our results.

## Results

A total of 8436 records were identified and, after deduplication, 4929 titles and abstracts were screened for inclusion (see Fig 1). After full-text screening of 303 articles, 33 articles representing 27 studies that represented foster children's views on foster care (and specifically their views on processes detailed in the inclusion criteria—removal from home, foster family processes, and placement breakdown) were included. From within this set of articles we found 25 articles [23, 40–63] representing 22 studies that focused on themes of participation. Details about participant characteristics are found in S3 Table, as well as discussed further below.

### Methodological quality

The methodological quality of the studies varied; the total score percentages for each article (total possible score was 20 'yeses') are reported in Table 2. Within most studies, authors outlined their strategies to ensure ethical issues had been taken into consideration (question 5), most authors had some strategies to ensure credibility of findings (question 6), and most studies addressed issues of transferability by discussing participant characteristics in sufficient depth to consider applicability to other contexts (question 7). Authors also tended to report

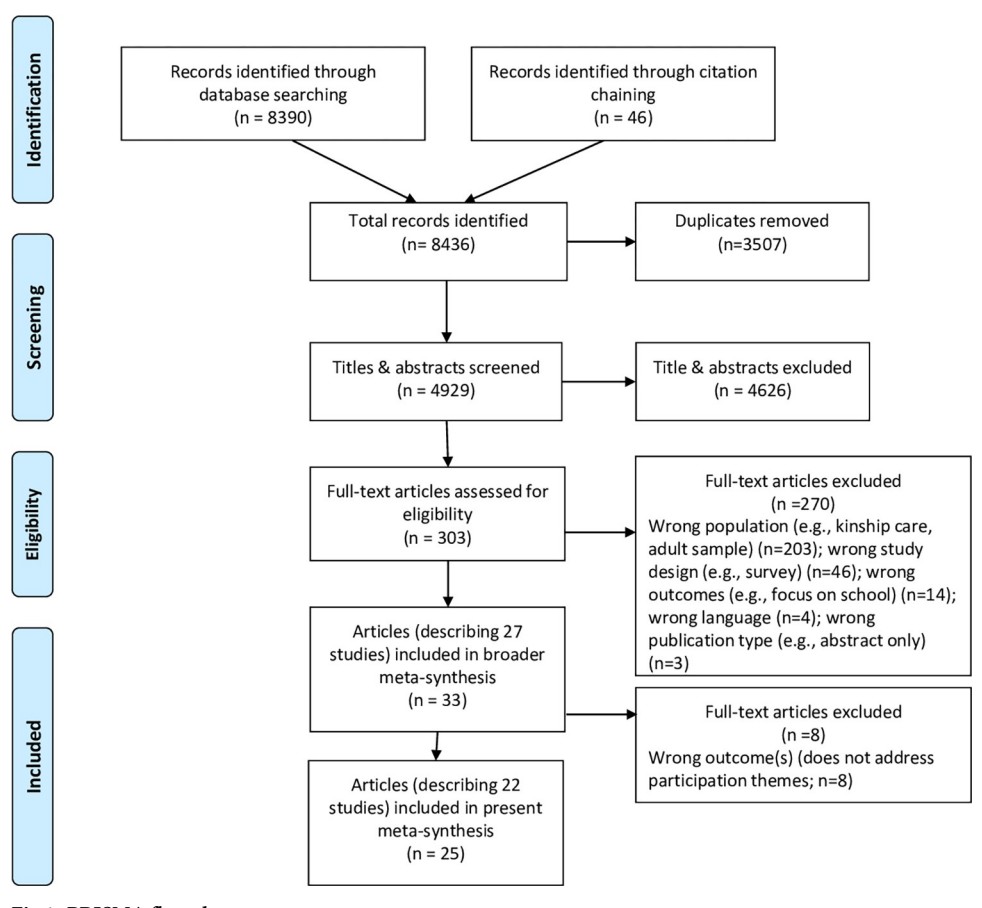

**Fig 1. PRISMA flow chart.**

clearly how data were collected (question 14) and to give explicit details about these methods (question 16). Other study reporting aspects related to 'consistency' (whether or not the reader can follow the study design decision trail) were mentioned less often by authors. For example, most study authors did not justify the location where interviews took place (question 13). Several studies did not give sufficient detail to understand their data analysis process (question 18) or sufficient details to interpret the presentation of their findings (question 19). About half of the studies addressed issues related to neutrality (question 20).

It is notable that some mixed-methods and multi-population (e.g., children, foster carers, social workers) studies ranked lower in methodological quality [47, 48], likely in part because of limited space to discuss multiple methods. Ethical concerns were a clear concern to researchers (question 5), yet concrete solutions to ethical concerns were not always apparent. For example, Winter [52] investigated younger children's perceptions and argued that their perspective is missing from research on foster care children.

As was discussed above, individual study quality did not affect the inclusion of children's quotes. Rather, the study quality illuminates the difficulties of research with vulnerable children as well as opportunities for improvement in study reporting.

## Participant and study characteristics

The studies took place in 11 high-income countries (see S3 Table for participant and study characteristics). Seven studies were conducted in England; three in the United States; two in

**Table 2. Quality appraisal scores across included studies.**

| Study ID | Critical appraisal questions[a] and answers[b] | | | | | | | | | | | | | | | | | | | | Counts (Yes) |
| | 1 | 2 | 3 | 4 | 5 | 6 | 7 | 8 | 9 | 10 | 11 | 12 | 13 | 14 | 15 | 16 | 17 | 18 | 19 | 20 | (n, %) |
|---|---|---|---|---|---|---|---|---|---|---|---|---|---|---|---|---|---|---|---|---|---|
| Carr 2017 | Y | Y | Y | Y | Y | Y | Y | Y | Y | Y | Y | Y | Y | Y | Y | Y | Y | Y | Y | N | 19 (95) |
| Mitchell 2010 | Y | Y | Y | Y | Y | Y | Y | Y | Y | Y | Y | N | Y | Y | Y | Y | Y | Y | Y | Y | 19 (95) |
| Whiting 2003 | Y | Y | Y | Y | Y | Y | Y | Y | Y | Y | Y | Y | U | Y | Y | Y | Y | Y | Y | Y | 19 (95) |
| Rostill-Brookes 2011 | Y | Y | Y | Y | Y | Y | Y | Y | Y | Y | N | N | Y | Y | Y | Y | Y | Y | Y | Y | 18 (90) |
| Daly 2009 | Y | Y | Y | Y | Y | Y | Y | Y | Y | Y | N | N | Y | Y | Y | Y | Y | N | Y | Y | 17 (85) |
| Rogers 2018 | Y | Y | N | N | Y | Y | Y | Y | Y | N | Y | Y | Y | Y | Y | Y | Y | Y | Y | Y | 17 (85) |
| Bogolub 2008 | Y | Y | U | N | Y | Y | Y | Y | Y | Y | Y | Y | Y | Y | N | Y | Y | Y | N | Y | 16 (80) |
| Madigan 2013 | Y | Y | Y | Y | Y | Y | Y | Y | Y | Y | N | N | N | Y | N | Y | Y | Y | Y | Y | 16 (80) |
| Skoog 2015 | Y | Y | Y | Y | Y | Y | Y | Y | Y | Y | Y | Y | N | Y | N | Y | Y | Y | N | N | 16 (80) |
| Degener 2020 | Y | Y | U | N | Y | Y | Y | Y | Y | Y | Y | N | Y | Y | Y | N | Y | Y | Y | N | 15 (75) |
| Ponciano 2013 | Y | Y | Y | Y | U | Y | Y | Y | U | Y | U | N | Y | Y | Y | Y | Y | Y | Y | Y | 15 (75) |
| Singer 2004 | Y | Y | U | Y | Y | Y | Y | Y | U | Y | Y | Y | N | Y | Y | Y | Y | Y | Y | U | 15 (75) |
| Winter 2010 | Y | Y | Y | N | Y | Y | Y | Y | N | Y | Y | N | Y | Y | Y | N | Y | N | Y | Y | 15 (75) |
| Wissö 2019 | Y | Y | U | N | Y | Y | Y | Y | Y | Y | Y | Y | Y | Y | Y | Y | U | N | N | N | 15 (75) |
| Goodyer 2016 | Y | Y | U | N | Y | Y | Y | Y | Y | Y | Y | Y | Y | Y | Y | N | N | N | Y | N | 14 (70) |
| Mosek 2004 | Y | Y | N | N | Y | Y | Y | Y | Y | Y | Y | N | Y | Y | N | Y | Y | Y | Y | N | 14 (70) |
| Polkki 2012 | Y | Y | U | N | Y | Y | Y | Y | Y | Y | Y | N | N | Y | Y | Y | Y | Y | Y | N | 14 (70) |
| Munro 2001 | Y | Y | U | N | Y | U | Y | Y | U | Y | Y | Y | N | Y | Y | Y | Y | N | N | Y | 13 (65) |
| Warming 2006 | Y | Y | Y | Y | Y | Y | Y | Y | U | U | U | Y | N | Y | Y | Y | N | N | N | Y | 13 (65) |
| Pert 2017 | Y | Y | U | N | Y | Y | Y | Y | Y | Y | Y | N | N | Y | Y | U | Y | N | Y | N | 12 (60) |
| Morrison 2011 | Y | Y | U | N | Y | Y | Y | Y | Y | N | N | N | N | Y | N | N | Y | Y | Y | N | 11 (55) |
| Dansey 2018 | Y | Y | U | N | Y | Y | Y | U | Y | U | N | N | N | Y | N | N | Y | Y | Y | N | 10 (50) |
| **Total (Yes)** | 22 | 22 | 10 | 9 | 21 | 21 | 22 | 21 | 16 | 19 | 12 | 11 | 8 | 22 | 16 | 19 | 20 | 15 | 16 | 11 | |

[a]Concepts addressed per CASP question (see full questions in example CASP form): 1) Research interprets actions and/or subjective experiences?; 2) Qualitative research the right methodology?; 3) Research design appropriate?; 4) Researcher justified research design?; 5) Researcher used 2+ strategies to address ethical issues?; 6) Researcher used 1+ strategies to establish credibility; 7) Researcher used strategies to establish transferability?; 8) Researcher used 1+ strategies to establish research purpose?; 9) Was the recruitment strategy appropriate to the research aim?; 10) Did the researcher explain how participants were selected?; 11) Did the researcher explain why the participants they selected were the most appropriate?; 12) Were there any discussions around recruitment?; 13) Was the setting for data collection was justified?; 15) Did the researcher justify the methods chosen?; 16) Did the researcher make the methods explicit?; 17) Is the form of the data clear?; 18) Did the researcher explain how the data were reduced or transformed for analysis?; 19) Did the researcher discuss their interpretation and presentation of their findings?; 20) Did the researcher use 1+ strategies to ensure neutrality?
[b]Possible answers for CASP questions include: Yes (Y), No (N), or Unsure (U).

Canada, the Netherlands, and Sweden; and one each in Australia, Denmark, Finland, Israel, Northern Ireland, and Scotland. Studies have increased somewhat over time: 4 from 2000–2004, 3 from 2005–2009, 7 from 2010–2014, 7 from 2015–2019, and 1 published in 2020. It is notable that 5 out of the 7 studies published from 2015–2019 were based in England.

These studies summarized the views of 376 children. Of the studies that indicated gender or sex of participants, there were 174 females/girls and 164 males/boys. Eleven studies discussed the race, ethnicity, or cultural background of children: 42 as Black or African American, 8 as Black British, 34 identified as White British, 17 as White, 24 as Native Dutch, 18 as migrant children, 4 as dual heritage (White British and Caribbean), 4 as "biracial" or "mixed", 5 as "minority ethnic background", 2 as Latino, and an unspecified number of children from one study [54] identified as Aboriginal, Torres Strait Islander, and Australian South Sea Islander. One study [61] that focused on the ethnic identity of adolescent foster children included

children from a variety of ethnic backgrounds (Moroccan and Dutch (1), Turkish and Dutch (1), Caribbean and Dutch (1), Surinamese and Turkish (1), Moroccan (4), Surinamese (1), Caribbean (5), East-African (5), Brazilian (1)).

Children's age ranges varied. Most studies sampled school-aged children (ages 6 to 12) and adolescents (ages 13 to 17) [23, 41, 42, 44, 45, 47, 49, 50, 54–56, 60, 61] and some sampled primarily school-aged children [40, 48, 57, 58] or adolescents [43, 46, 62, 63]. Only two studies investigated children under seven [52, 56].

Most studies included 100 percent foster care children, while three studies [45, 52, 60] included a few children from other settings, such as at home, residential care, or kinship care. Most sampled children had been in foster care across a wide range of duration, such as 1 to 14 years [47], 4 to 12 years [49], 9 months to 9 years [42], or 6 months to 5 years [52]. Two studies investigated children's experiences shortly after being placed in foster care, resulting in shorter timeframes—6 to 36 months [50] and 1 to 5 months [55]. Eight studies discussed number of placements experienced by children [40, 41, 45, 47, 49, 52, 57, 58]; included children experienced a wide range of placements, such as 1 to 6 [40], 1 to 8 [58], or 1 to 11 placements [45].

## Theoretical frameworks

Several of the studies [23, 44, 47, 52, 54] cited the 1989 United Nations Convention on the Rights of the Child or associated legislation (e.g., Children Act 1989, the main source of child welfare law for England and Wales) to justify the necessity of investigating children's perspectives. Common theoretical frameworks used by study authors across time to frame their study or findings were theories of attachment [40, 41, 43, 48, 49, 63], child development [41, 45, 46, 49, 50, 58, 63], child well-being [40, 41, 45, 46, 49, 50, 52, 56] and identity development [23, 40, 47, 52, 54, 56, 61, 63, 64]. Another influential discourse was the "best interests of the child" which was referenced by eight studies [23, 46–48, 50, 56, 60, 62], two from a critical perspective [23, 60]. The two studies [23, 60] that were critical of the bests interests of the child discourse considered the impact of power imbalances between adults and children.

## Facets of participation

Children's perspectives on participation in child welfare are found in S4 and S5 Tables and summarized in Table 3 below. Children discussed a number of 'facets' of participation, including 1) what children are (not) told by adults; 2) how children are (not) prepared by adults; 3) what children do (not) know; 4) what adults (do not) ask children; 5) how children (do not) talk; 6) how adults (do not) listen to children; and 7) how children (do not) participate and decide (see S4 Table). Each facet of participation discussed by children has positive/negative possibilities, such as being told versus not being told. In many but not all cases, "positive" themes were described as positive experiences by children (e.g., children want to be told about aspects of foster care and described having negative experiences if they were not told), but there were exceptions (e.g., being informed about going into foster care by a taxi driver was not a positive experience for a child).

The separation of these facets of participation is artificial, as children would often combine these facets depending on their needs. For example, they might express the desire to receive information about rules in a foster care home so they could participate in decision-making.

Studies inconsistently labeled individual quotes by age and gender and rarely labeled quotes by other factors (e.g., ethnicity, placement status), so analysis at this level was unfortunately not possible. In spite of the wide variety of contexts from which sampled children were speaking from (11 different high-income countries, different genders, ages, placement lengths, etc.), there seemed to be similarities in terms of how children discussed participation. For example,

**Table 3. Strategies for enhancing foster children's participation.**

| Facets of participation discussed by children | Strategies for enhancing foster children's participation | Facilitators of participation discussed by children |
|---|---|---|
| **Adults[a] telling** | • Adults give children honest, developmentally appropriate information about foster care | *Honesty, developmental appropriateness* |
| | • Adults explain limits of confidentiality to children | *Confidentiality* |
| | • Adults give information to children before decisions are made or changes occur | *Timing* |
| | • Adults notice and communicate to children about things they are doing well | *Appreciation* |
| **Adults[a] preparing** | • dults prepare children for any foster care-related changes *before* these changes take place; changes are not rushed | *Timing* |
| | • Adults facilitate children's introductions into foster families in ways that are developmentally appropriate and support relationship connection | *Developmental appropriateness, connection* |
| | • Adults offer children meaningful choices related to decisions that affect their lives, including attention to children's race, ethnicity, and culture | *Choices, power, belonging* |
| **Adults[a] asking** | • Adults consider the safety of children when asking them about their lives (e.g., they do not ask safety-related questions in front of parents) | *Power* |
| | • Adults ask children about important aspects of their lives and changes they would like to make | *Inclusivity, power* |
| | • Adults ask children about things that are important to them (thoughts, feelings, wishes, needs, favourites) | *Thoughts, feelings, behaviours, and wishes, connection* |
| **Children talking** | • Adults support children to understand their thoughts, feelings, behaviours, needs, and wishes so they can talk about them if they want | *Thoughts, feelings, behaviours, and wishes, wishes, power* |
| | • Adults share power by creating opportunities for children to share | *Power, inclusivity* |
| **Adults[a] listening** | • Adults demonstrate appreciation for children's sharing | *Appreciation* |
| | • Adults actively try to understand children's thoughts, feelings, needs, behaviours, and wishes | *Understanding, thoughts, feelings, behaviours, and wishes* |
| **Children participating and deciding** | • Adults create meaningful opportunities for children to participate and decide; these opportunities take into consideration children's preferences (e.g., time of meetings) | *Inclusivity, power* |
| | • Adults share power with children; they do not intentionally exclude children from conversations of importance to their lives (e.g., care planning) | *Power* |
| | • Adults act with integrity and are careful about what they promise they can do | *Honesty* |
| | • Adults effectively work to counteract children's feelings of helplessness, exclusion, unfairness, and lack of trust | *Self-efficacy, inclusivity, fairness, trust* |
| | • Adults facilitate children's access to technology, especially when meaningful for their participation and connection | *Access to technology, connection* |

[a]Ideally adults are not strangers to children, as reflected in children's preferences for strong relationships with meaningful adults.

regarding facets of participation, both school-aged children and adolescents from a variety of high-income countries contributed to each theme. In addition, all 11 countries (Australia, Canada, Denmark, England, Finland, Israel, Netherlands, Northern Ireland, Scotland, Sweden, and United States) contributed illustrative quotes to the facet "children talk," while only four countries (Canada, England, Scotland, United States) contributed illustrative quotes to the facet "children are prepared" (see S4 Table).

## Facilitators and barriers to participation

Within the facets of participation discussed above, children expressed ways that their participation was blocked or enhanced (see S5 Table). These facilitators and barriers to participation seemed to relate to the following themes:

• Children's individual qualities (thoughts, feelings, behaviours, wishes, self-efficacy);

- Aspects of the information they received, such as who told the child about foster care (informants), what children were told (developmental appropriateness, choices), when they were told (timing), and how they communicated with meaningful adults (access to technological means); and

- Qualities of relationships (or of adults) that children found important when participating, including appreciation, availability, belonging, connection, confidentiality, inclusivity, honesty, fairness, shared power, understanding, and trust.

Possibly related to the higher number of studies coming from England, it is notable that foster children from England contributed to all facilitators and barriers to participation themes, whereas foster children from Israel only contributed to one theme (thoughts, feelings, wishes, and behaviours) (see S5 Table). Although not necessarily a reflection of the importance of each theme, it is notable that foster children from the most countries contributed to themes about how their participation was facilitated or impeded by a) their thoughts, feelings, wishes, and behaviours (Canada, England, Finland, Israel, Netherlands, Sweden, United States) or b) different aspects of power (Denmark, England, Finland, Netherlands, Northern Ireland, Scotland, Sweden, United States) (see S5 Table). The qualities of relationships commented on by children from the most countries include appreciation (5 countries) and understanding (7 countries), whereas only two countries contributed to the qualities of availability, inclusivity, and trust (see S5 Table).

### Considering facets of participation with barriers and facilitators of participation

Table 3 summarizes elements of participation that were important to foster children for each facet. In most cases, themes are presented in the positive (i.e., what children want adults to do).

## Discussion

This meta-synthesis of qualitative research summarizes foster children's (non-) participation in the removal from home, foster family processes, and placement breakdown. The findings suggest that much work needs to be done to improve children's participation in foster care. Foster children described many facets of participation (see S4 Table), including what they (do not) know about aspects of foster care because they were told or prepared by adults; how adults asked (or didn't ask) about aspects of children's lives of importance to them; how children spoke about their lives and if adults listened; and how children have (not) participated in or made decisions about aspects of their lives. While there are some children who were prepared for changes while in foster care, many children described not knowing about a variety of aspects of their lives—why they were removed from their home, when they will get to see their family again, who their foster family will be, and so on (see S4 Table). Many children described not knowing and not being prepared as traumatic. For example, some children reported that removal from home felt like "being kidnapped" (see S4 Table). This finding is consistent with other research which has found that some children describe their primary experience of trauma as being related to removal from their home [65].

The results of this meta-synthesis differ from other reviews on participation in child welfare processes [16, 24–28, 66], in terms of its deliberate focus on children's voices and ways of understanding and describing participation. Nevertheless, the findings of this review have similar conclusions as the other reviews, in terms of the acknowledgment of the importance of children's participation in child welfare processes within the context of strong relationships, as

well as the stressful and potentially traumatic nature of children's contact with child welfare. As noted in the Introduction, the findings of this meta-synthesis will be useful to both policy makers and practitioners. For example, policy makers who want to increase children's awareness of foster care processes may find the particular areas where children expressed 'not knowing' of importance, such as their personal history prior to foster care; what placement or foster care means; what their foster family will be like; why they became involved with child protection services, and so on (see S4 Table). They may also seek to improve areas of children's experiences where they expressed considerable dissatisfaction, such as their transitions into care. Child welfare managers may want to consider areas of their practice where they are well aligned or not well aligned with children's wishes, such as giving children information before decisions are made, changes occur or strategies to share power with children (see Table 3). They may also want to consider the qualities of professionals that children value (e.g., availability, honesty, fairness) and how these values are exemplified (or not) in their service delivery (see S5 Table).

## Foster care reform: Considering foster children's perspectives

Foster children identified many facilitators and barriers to their participation, which suggest avenues for meaningful changes to improve participation processes for foster children (see S5 Table). These barriers and facilitators can be helpfully organized according to the sociological model, including attention to macro factors (laws, policy, social norms), exo- and meso-system factors (social systems, including available community resources, and relationships among organizations and institutions), microsystem factors (formal and informal relationships), and individual factors (knowledge, attitudes, skills, etc.). Doing so draws attention to the importance of children's rights to know, be involved, and be prepared in all aspects of foster care, as well as their attention to the importance of strong relationships with meaningful adults. Below we summarize foster children's perspectives across socioecological levels.

**Macro-system considerations.** The findings of this meta-synthesis suggest that all aspects of foster care should attend to foster children's rights and consider their unique experiences. This corresponds with the United Nations Convention on the Rights of the Child, which was cited by several authors in this meta-synthesis [23, 44, 47, 52, 54] as well as many other authors who have considered the rights of foster children [67–69]. According to foster children, values informing all aspects of foster care should be: children receiving information they understand and aspects of care being of relevance and appropriate to their age and stage (developmental appropriateness); adults sharing power with children (e.g., moving from *listening* to *acting upon* foster children's sharing); attention to culture, race, ethnicity, and diversity (e.g., same-race, culture, or ethnicity placements when desired by child); serious attention to how foster care processes can be child-centred (e.g., planning care reviews at a time that considers the child's needs, such as building relationships with friends after school); strength-based processes (e.g., noticing what children do well); community-based options (e.g., most children do not want to move neighbourhoods, schools, or communities); and services that prioritize relationships with trustworthy adults (e.g., foster children wanted social workers who listened to them, understood them, were helpful and available).

At the policy level, this meta-synthesis also draws attention to three important children's rights: children's right to know, children's right to be involved, and children's right to be prepared. Foster children wanted to know what happened in the past (e.g., why they needed to be placed in foster care), what is happening to them now (e.g., the meaning and purpose of foster care, where they will be living, the family with whom they will be placed), and what these processes mean for them in the present, near and distant future. In addition to the right to know,

foster children should be prepared about all major changes in their lives, including removal from the home but also any changes in placement. Preparations should take into account children's unique material, social and emotional needs [42, 49]. Foster children also have the right to be involved in all aspects of care, including but not limited to giving them the opportunity to ask questions, voice concerns, or be included in decision-making. As discussed above regarding consideration of values, these opportunities are unique to each child (e.g., some children prefer to be passive listeners), including attention to their age and stage.

A key research and policy consideration, therefore, is how foster care can be meaningfully reorganized so that it is as child friendly as possible. There is a sentiment that foster care is uniformly negative and should be avoided at all costs, which minimizes creative thinking about how foster care can be better organized for those children for whom it is necessary. In a recent policy report, Font and Gershoff [70] argued that governments must shift from quantitative performance outcomes to children-centred outcomes that focus on children's needs, safety and care. That so many children struggled with removal from home processes suggest that it would be meaningful to consider the transition into care as a distinct phase of service delivery. In doing so, transition into care should be informed by children's experiences, best practices, guidelines, and standards [50, 58]. The absence of research on strategies to improve transition into care compared to other areas of healthcare is notable. For example, what might it look like to have child-life specialists involved with preparing children for foster care, similar in principle to how children are prepared for pediatric care in healthcare settings [71–74]?

**Meso- and exo-system considerations.** Children did not draw attention to community/institutional factors affecting them, beyond noticing the impacts of system-level stress, such as noticing that social workers were stressed or unavailable or that their foster home did not meet their needs. Although foster children did not discuss coordination of services specifically, they did discuss the belief that they could remedy placement breakdown if knowledge about difficulties was provided to them in advance (e.g., preventative dialogue across social workers and foster carers and with foster children).

Some children's struggles adjusting to new foster families suggests that children should have access to foster care homes that are safe, but also that they attend to their material, social, and emotional needs [40, 49]. This could involve matching "children to foster homes where they are most likely to thrive" [70]. Children found social workers who supported them especially important; social workers can play an important role in negotiating and advocating with new foster carers on behalf of children's tastes, preferences, and routines [49, 52]. Children's unique experiences can be meaningfully incorporated into all aspects of social work practices, including child assessments, decision-making, planning, and advocacy [44, 49, 52, 58]. For example, social connections and attachments were a key concern raised by foster children. As such, children's social networks and attachments (including people, pets, places, and personal belongings) should be carefully assessed and strengthened in care planning [41, 43, 45, 50, 52, 55, 58].

**Relational and skill-based (individual) considerations.** Foster children's participation-related concerns were primarily focused on factors in their relationships, such as if they were appreciated when they shared their views; if they were included in decisions or processes of importance to their lives; or if they felt like they belonged. Children also described aspects of power in their relationships, such as if adults were oppressive or if they used their power to help them. With this in mind it seems important to consider how social work and foster care homes could be organized in such a way as to prioritize meaningful, respectful relationships with children in care [51, 52].

In terms of foster carers, it is clear that meeting the needs of foster children increasingly requires advanced knowledge of attachment, development, and trauma, but also many aspects

of caring by foster carers defies attempts to 'professionalize' it (i.e., it involves innate or developed strengths related to caring well) [75–77]. Children's attention to relationships with social workers also suggests that it would be helpful if child welfare was organized in such a way that it enables social workers to meet with children many times and spend enough time with them to develop a relationship [47]. Many social service texts are reprioritizing the importance of relationships [78], however, there still continues to be an emphasis on administrative processes, such as measurement, rather than relational processes [79].

Finally, in terms of individual considerations, foster children's recognition that good management of their feelings was a facilitator to their participation suggests that social workers and foster carers also need skills to support foster children with their feelings, especially when they blame themselves about their circumstances or when they hide their feelings and needs [52, 57, 58].

## Strengths, limitations and future research

The strengths of this meta-synthesis include the use of a systematic search, clear a priori study inclusion and exclusion criteria, use of an established study appraisal checklist, and transparent and reproducible methods for analysis. This review reflects the expressed views of foster children's perspectives on participation in foster family processes, and placement considerations. Complementary reviews are needed to understand children's perspectives in other out-of-home care settings and foster children's perspectives on other aspects of their lives (e.g., health and mental health support). An additional limitation of this paper is that we included only peer-reviewed journal articles and should acknowledge that there are several governmental reports that have sought foster care children's perspectives. The lack of studies in this synthesis from low- and middle-income country settings suggests an increased need to invest in research in these settings, as well as in examining differences in the organization of care in these settings. It is important to note that some studies had lower quality appraisal scores, but no studies were excluded based on quality scores so as not to exclude relevant quotes from children. Lower methodological scores highlight the need for better reporting of qualitative research and the continued need to conduct ethical research with vulnerable children. As studies inconsistently labeled individual quotes (the unit of analysis) with certain demographic details (e.g., age, gender) and very rarely labeled individual quotes with other demographic information (e.g., race/ethnicity, placement information), it was not possible for us to determine how these factors influenced study findings. Future quantitative research investigating children's participation may be able to discern if or how participation differs for children depending on their demographic factors or other information (e.g., country, child welfare context). In addition, while the participation themes were remarkably similar across the 11 high-income countries included in this review, we can hypothesize that the context of each country's foster care system would impact the participation of children. For example, the fact that so many published studies were from England suggests a shift in policy to emphasize children's voices, a point that was discussed by the authors of one study [42]. Future research investigating the impact of participation would benefit from clear explanations of the contextual factors influencing children's participation (e.g., details about the country's foster care system at the time of data collection and any relevant legislation or child welfare initiatives that impact children's participation). Future research is also needed to address what concrete, positive changes can be made to improve foster children's experiences of participation across their experiences in care.

## Conclusion

While adults have a responsibility to protect children from harm, this responsibility should not override foster children's right to participation. In this meta-synthesis, foster children

identified many ways they were or were not able to participate in aspects of child welfare processes. No one way of participating is better than another, as some children more clearly desired passive listening roles and others desired active roles in decision-making. However, meaningful adults in foster children's lives have a responsibility to strengthen the emphasis on children's needs and voices in child welfare processes.

## Supporting information

**S1 Table. PRISMA and ENTREQ checklists.**
(DOCX)

**S2 Table. Modified CASP critical appraisal form.**
(DOCX)

**S3 Table. Participant and study characteristics.**
(DOCX)

**S4 Table. Facets of participation.**
(DOCX)

**S5 Table. Facilitators and barriers of foster children's participation.**
(DOCX)

**S1 File. Example search.**
(DOCX)

## Author Contributions

**Conceptualization:** Jill R. McTavish, Harriet L. MacMillan.

**Formal analysis:** Jill R. McTavish, Christine McKee, Harriet L. MacMillan.

**Methodology:** Jill R. McTavish, Harriet L. MacMillan.

**Project administration:** Jill R. McTavish.

**Resources:** Harriet L. MacMillan.

**Supervision:** Harriet L. MacMillan.

**Writing – original draft:** Jill R. McTavish.

**Writing – review & editing:** Jill R. McTavish, Christine McKee, Harriet L. MacMillan.

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
