## [Decision Letter · Decision Letter 0]

7 Jun 2022

PONE-D-21-12708

Foster children’s perspectives on participation in child welfare processes: A meta-synthesis of qualitative studies

PLOS ONE

Dear Dr. McTavish,

We are hear Jill!!! And great news from the peer review process, only minor revisions.  Thank you for submitting your manuscript to PLOS ONE. After careful consideration, we feel that it has merit but does not fully meet PLOS ONE’s publication criteria as it currently stands. Therefore, we invite you to submit a revised version of the manuscript that addresses the minor points raised during the review process.

Please submit your revised manuscript by July 22, 2022.  If you will need more time than this to complete your revisions, please reply to this message or contact the journal office at plosone@plos.org. Please include the following items when submitting your revised manuscript:

We look forward to receiving your revised manuscript.  And FINALLY :)

Kind regards,

Melissa Sharer

Academic Editor

PLOS ONE

Journal Requirements:

2. Please attach a Supplemental file of the results of the CASP quality assessment for each individual study assessed, reporting the outcome for each individual criteria considered.

Reviewers' comments:

Reviewer's Responses to Questions

**Comments to the Author**

1. Is the manuscript technically sound, and do the data support the conclusions?

Reviewer #1: Yes

Reviewer #2: Yes

2. Has the statistical analysis been performed appropriately and rigorously? 

Reviewer #1: Yes

Reviewer #2: Yes

3. Have the authors made all data underlying the findings in their manuscript fully available?

Reviewer #1: Yes

Reviewer #2: Yes

4. Is the manuscript presented in an intelligible fashion and written in standard English?

Reviewer #1: Yes

Reviewer #2: Yes

5. Review Comments to the Author

Reviewer #1: This is a well written manuscript covering an important topic. The authors should be congratulated for their thorough and thoughtful work. I am providing some suggestions and recommendations to further strengthen the paper.

Reviewer #2: This is to me a high quality meta-synthesis of qualitative research about the experiences of foster children, a highly vulnerable population. This paper is well written. It is novel in that it appears no meta-synthesis of foster children's experiences in the qualitative literature has been done.

6. PLOS authors have the option to publish the peer review history of their article (what does this mean?). If published, this will include your full peer review and any attached files.

Reviewer #1: No

Reviewer #2: No

---

## [Author Response · Author response to Decision Letter 0]

4 Jul 2022

This information is copied from the Reponses to Reviewers File: 

We would like to thank the Editors and Reviewers for their thoughtful and constructive feedback, which we believe has improved the manuscript. Editor and Reviewer responses are found in bold below, with our responses underneath. 

Responses to editor:

• We have carefully gone through these documents to ensure they match the style requirements. 

2. Please attach a Supplemental file of the results of the CASP quality assessment for each individual study assessed, reporting the outcome for each individual criteria considered.

• We have reported the outcome for each of the criteria in Table 2 and have provided a modified CASP form in S3 as an example.

• Our excel file of data coding was listed as ‘available from author’. There are no ethical/legal restrictions to sharing this file. 

• The excel files of coding was submitted to Dryad: https://doi.org/10.5061/dryad.8pk0p2nqs. Data remains unpublished during the review process and can be accessed here: https://datadryad.org/stash/share/DSWnUl_i6XiOAmsQmehksryRzaq3JI7su69eECjBbzs. 

• The minimal data set is now available in the manuscript or in dryad.

• We have included captions for the Supporting Information files, as requested. 

• We have not removed any references. We have added some references to respond to Reviewer #1’s request (for example, to contextualize the findings within the context of recently published reviews). 

Reviewers' comments:

Reviewer's Responses to Questions

5. Review Comments to the Author

Reviewer #1: This is a well written manuscript covering an important topic. The authors should be congratulated for their thorough and thoughtful work. I am providing some suggestions and recommendations to further strengthen the paper.

• Thank you for your positive comments about our manuscript.

1. Line 37 page 3. Reference 11 is noted twice in the same reference string.

• Thank you, we have deleted this double reference.

2. Row 43, page 3. It would be helpful to define what “child welfare processes” consist of in this context, particularly given this reviews covers a wide variety of contexts. Do these vary across country settings and how did the authors define it?

• This is a good point to clarify. We have generally been quite broad in our understanding of child welfare (called different things by different countries, such as ‘children’s social services’ or ‘children’s aid’). We added these sentences in the Introduction to clarify: “Child welfare processes and level of service response vary by country (20). For example, most countries allow for voluntary reporting of child maltreatment (with considerable variability in mandatory reporting requirements), require reports to be investigated within a set time period, and also require that some sort of services are provided, such as services for parents (e.g., substance use treatment), for children (e.g., therapy programs), or general services (e.g., universal free medical care) (20).”

• You’ll notice from our Table 1 exclusion criteria that we focused specifically on child welfare processes and not on the types of community services to which child welfare might refer children to (e.g., mental health support). We also excluded some important child welfare processes (e.g., transitions from care) as a lot of intensive work in this area has been done, to which readers can refer. We added these sentences in the methods to clarify: “The inclusion criteria for the larger review are as follows: (1) English-language, (2) primary studies that used a qualitative design; (3) published articles; (4) investigations of children’s self-reported experiences of foster care, which could include: removal from home, foster family processes, or placement breakdown (see Table 1 for full inclusion criteria of the broader review). It could also include children’s self-reported experiences of formal and informal relationships while in foster care, including with social workers, foster carers, peers, or biological family members (siblings, parents, grandparents, aunts/uncles, cousins). The broader review excluded some themes that have been addressed by recent reviews, such as children’s self-reported experiences with violence itself (e.g., barriers to disclosure (33)), transitions out of foster care (34), and experiences at school (35).”

3. The final paragraph in the introduction can be strengthened by adding more detail about other systematic reviews that have been conducted within this area, and how this systematic review differs/adds to that knowledge base. 

• Our review is unique in that we prioritized children’s voices in how our results were organized. This is useful for ethical reasons (children’s voices matter) and for practical reasons (to better understand and respond to children’s preferences around participation). We have clarified this in the Introduction and Discussion. 

4. Also in the final paragraph of the introduction, it seems clear that the aim is to summarize children’s perspectives. However, the reader is left to wonder, to what purpose? Who can use the information from this review and how can they use it? How is this information applicable within a practice or policy setting? 

• We hope this review will be useful for policy makers, who are increasingly requested to consider children’s voices when making decisions about their lives, and practitioners, who seem to struggle with the practicalities of how to include children’s voices. We have added this to the Introduction: “The findings of this meta-synthesis will be useful to policy makers, who are increasingly requested to incorporate children’s voices into decision-making (16,24,25), and practitioners, who need to consider the nuances of when and how to include children’s voices in practice and decision-making (26). The inclusion of children’s voices has a number of benefits to children, practitioners, and policy makers, such as affording children their inherent rights to participate, empowering children and reducing their confusion regarding services, and improving service delivery through more tailored, responsive services (24,26,27). As the authors of one review note, children’s “participation has both intrinsic (dignity and self-worth in terms of expressing views to influence decisions about their lives) and instrumental (policy and better outcomes for children in terms of supportive relationships with their workers and positive experiences at school and in CPS [children protection services]) value” (24).This meta-synthesis adds to an important increasing focus in the literature on children’s participation across all aspects of child welfare processes (24,26–28), but with an explicit focus on how foster children speak about their participation.” And this to the Discussion: “As noted in the Introduction, the findings of this meta-synthesis will be useful to both policy makers and practitioners. For example, policy makers who want to increase children’s awareness of foster care processes may find the particular areas where children expressed ‘not knowing’ of importance, such as their personal history prior to foster care; what placement or foster care means; what their foster family will be like; why they became involved with child protection services, and so on (see S5). They may also seek to improve areas of children’s experiences where they expressed considerable dissatisfaction, such as their transitions into care. Child welfare managers may want to consider areas of their practice where they are well aligned or not well aligned with children’s wishes, such as giving children information before decisions are made, changes occur or strategies to share power with children (see Table 3). They may also want to consider the qualities of professionals that children value (e.g., availability, honesty, fairness) and how these values are exemplified (or not) in their service delivery (see S6).”

5. Search strategy, line 74. Is there a reason why key informant interviews were not included in the strategy?

• We think that key child informant interviews would be included if they were available? We included index terms (“Interviews/”) and keywords (“interview*”) which should capture all types of interviews, including key informant interviews. In our experience, the inclusion of more search terms leads to less results as the results have to match the keywords (e.g., searching for “key informant interview*” would not catch “key child informant interviews*). But perhaps you mean other informants? Such as social workers? In which case this is a very relevant area of research that would be complementary to the findings of this review. 

6. Within the results section, there should be more information (beyond a supplementary table) regarding the CASP findings/methodological rigor of the included studies beyond inclusion of the table. What elements were weak that might have implications for the results?

• We have made a more comprehensive table of the CASP findings so that readers can review the overall quality of the studies and for each CASP question. We have also pulled our summary of the study results from the supplementary file into the Methods section, so that there is more information about the weak/strong study elements. As noted in the Methods section, we did not exclude any studies based on methodological rigour, as we felt this would exclude important child voices. We have, however, included a note in the Discussion section that some studies were methodologically weak and that this speaks to the need for better reporting of qualitative results and continued striving to conduct ethical research with vulnerable children: “It is important to note that some studies had lower quality appraisal scores, but no studies were excluded based on quality scores so as not to exclude relevant quotes from children. Lower methodological scores highlight the need for better reporting of qualitative research and the continued need to conduct ethical research with vulnerable children.”

7. Within the results section, findings are lumped mainly into a single table, and the discussion begins with new information that actually appears that it should belong in the results. It would be helpful to include more detailed information beyond the table within the results section, and also a more detailed analysis of findings across different contexts. For example, the participants are included from a variety of developed country settings around the world wherein the foster care system likely vastly differs. It would be helpful to provide more detailed information by either continent (or preferably) country-setting to identify if there are specific challenges/opportunities within individual contexts. 

• We gave careful consideration to how to make the results of the review accessible. Originally all of the text from Supplementary file 5 (Facets-the table and explanatory text below the table) and 6 (Facilitators and Barriers-the table and explanatory text below the table) was in the Results, which is over 4500 words of content. We have positioned this important work in the supplementary files and also have a summary table (now Table 3), so that the overall findings are more accessible to readers. We acknowledge that much of the important information is in the supplementary files and we try to draw readers attention to it in the Discussion, as you have noted. We have added more references to the supplemental files in the Discussion to strengthen the emphasis on important results; we indicate how interested readers can refer to the supplemental files for more details. 

• In addition, we added the countries contributing to each theme in the right-hand column of Supplemental file 5 and 6 and added text to explain these findings in the Results (see pages 16, lines 266-276 and pages 17, lines 289-299). This provides some information about “context” to the themes. It is of note that the participation-related themes were very consistent across variable countries and contexts. However, we agree that this would be a useful area of future exploration—to better understand how contextual factors impact participation. Unfortunately, studies did not provide enough detail about the context of the foster care system in their countries to comment on this point. For example, the fact that so many studies are published from England suggests a policy shift to increase children’s voices, but this was only alluded to by the authors of one of the English studies. As the child welfare system is constantly shifting to respond to layered (often reactive) policies and procedures, we were hesitant to speculate about the context of the foster care systems in the countries at the time of data collection. We have added this important point to the limitations of the findings: “In addition, while the participation themes were remarkedly similar across the 11 high-income countries included in this review, we can hypothesize that the context of each country’s foster care system would impact the participation of children. For example, the fact that so many published studies were from England suggests a shift in policy to emphasize children’s voices, a point that was discussed by the authors of one study (42). Future research investigating the impact of participation would benefit from clear explanations of the contextual factors influencing children’s participation (e.g., details about the country’s foster care system at the time of data collection and any relevant legislation or child welfare initiatives that impact children’s participation).”

8. In addition to the above, a more detailed understanding of how children experience the foster care system by race/ethnicity/minority status/age/gender would be very interesting. 

• Unfortunately, studies inconsistently labeled individual quotes with gender and age of the child and very rarely labeled individual quotes with other relevant factors (e.g., race/ethnicity/placement information). For this reason, we weren’t able to do this kind of in-depth analysis. This might be more possible with quantitative data. We have now included this as a limitation of our findings that deserves future attention: “As studies inconsistently labeled individual quotes (the unit of analysis) with certain demographic details (e.g., age, gender) and very rarely labeled individual quotes with other demographic information (e.g., race/ethnicity, placement information), it was not possible for us to determine how these factors influenced study findings. Future quantitative research investigating children’s participation may be able to discern if or how participation differs for children depending on their demographic factors or other information (e.g., country, child welfare context).”

Reviewer #2: This is to me a high quality meta-synthesis of qualitative research about the experiences of foster children, a highly vulnerable population. This paper is well written. It is novel in that it appears no meta-synthesis of foster children's experiences in the qualitative literature has been done.

• Thank you for your positive feedback. We hope the results are useful for policy makers and practitioners.

---

## [Decision Letter · Decision Letter 1]

13 Sep 2022

PONE-D-21-12708R1Foster children’s perspectives on participation in child welfare processes: A meta-synthesis of qualitative studiesPLOS ONE

Dear Dr. McTavish,

Thank you for submitting your manuscript to PLOS ONE. After careful consideration, we feel that it has merit but does not fully meet PLOS ONE’s publication criteria as it currently stands. Therefore, we invite you to submit a revised version of the manuscript that addresses the points raised during the review process.

A marked-up copy of your manuscript that highlights changes made to the original version. You should upload this as a separate file labeled 'Revised Manuscript with Track Changes'.An unmarked version of your revised paper without tracked changes. You should upload this as a separate file labeled 'Manuscript'.

We look forward to receiving your revised manuscript.

Kind regards,

Yann Benetreau

Staff Editor

PLOS ONE

Journal Requirements:

Additional Editor Comments (if provided):

I was told that you had submitted a revised manuscript to our editorial office, and I am issuing this decision so that you can submit your revision via Editorial Manager. Hopefully a final decision can be issued swiftly after resubmission.

Reviewers' comments:

Reviewer's Responses to Questions

**Comments to the Author**

1. If the authors have adequately addressed your comments raised in a previous round of review and you feel that this manuscript is now acceptable for publication, you may indicate that here to bypass the “Comments to the Author” section, enter your conflict of interest statement in the “Confidential to Editor” section, and submit your "Accept" recommendation.

Reviewer #1: All comments have been addressed

Reviewer #3: (No Response)

2. Is the manuscript technically sound, and do the data support the conclusions?

Reviewer #1: Yes

Reviewer #3: Yes

3. Has the statistical analysis been performed appropriately and rigorously? 

Reviewer #1: N/A

Reviewer #3: Yes

4. Have the authors made all data underlying the findings in their manuscript fully available?

Reviewer #1: Yes

Reviewer #3: Yes

5. Is the manuscript presented in an intelligible fashion and written in standard English?

Reviewer #1: Yes

Reviewer #3: Yes

6. Review Comments to the Author

Reviewer #1: The authors were responsive to my comments. This is a well written manuscript and is an important contribution to the literature.

Reviewer #3: I appreciate the authors’ explanation of why they were not able to address my comments previously. I also appreciate the response they have now made. As far as comment 1 is concerned, I am entirely happy. For comment 2, I would just point out that it’s the Children Act 1989 and the Children Act 2004, not the the Children’s Act, and also that the legislation only applies in England and Wales, not in the whole UK.

7. PLOS authors have the option to publish the peer review history of their article (what does this mean?). If published, this will include your full peer review and any attached files.

Reviewer #1: No

Reviewer #3: No

---

## [Author Response · Author response to Decision Letter 1]

15 Sep 2022

We are pleased that the Reviewers have found our revisions responsive to their comments and have submitted the updated manuscript to the Editorial Manager, as requested. Editor and Reviewer responses are found in bold below, with our responses underneath. 

Additional Editor Comments (if provided):

I was told that you had submitted a revised manuscript to our editorial office, and I am issuing this decision so that you can submit your revision via Editorial Manager. Hopefully a final decision can be issued swiftly after resubmission.

• We have submitted the revision via the Editorial Manager, with the slight change to the Children Act 1989, as per Reviewer #3’s suggestion.

Reviewers' comments:

Reviewer's Responses to Questions

Comments to the Author

1. If the authors have adequately addressed your comments raised in a previous round of review and you feel that this manuscript is now acceptable for publication, you may indicate that here to bypass the “Comments to the Author” section, enter your conflict of interest statement in the “Confidential to Editor” section, and submit your "Accept" recommendation.

Reviewer #1: All comments have been addressed

Reviewer #3: (No Response)

2. Is the manuscript technically sound, and do the data support the conclusions?

Reviewer #1: Yes

Reviewer #3: Yes

3. Has the statistical analysis been performed appropriately and rigorously?

Reviewer #1: N/A

Reviewer #3: Yes

4. Have the authors made all data underlying the findings in their manuscript fully available?

Reviewer #1: Yes

Reviewer #3: Yes

5. Is the manuscript presented in an intelligible fashion and written in standard English?

Reviewer #1: Yes

Reviewer #3: Yes

6. Review Comments to the Author

Reviewer #1: The authors were responsive to my comments. This is a well written manuscript and is an important contribution to the literature.

• We are pleased that Reviewer #1 found our revisions to be responsive to their helpful comments.

Reviewer #3: I appreciate the authors’ explanation of why they were not able to address my comments previously. I also appreciate the response they have now made. As far as comment 1 is concerned, I am entirely happy. For comment 2, I would just point out that it’s the Children Act 1989 and the Children Act 2004, not the the Children’s Act, and also that the legislation only applies in England and Wales, not in the whole UK.

• Thank you for identifying this error in our reference to the Children Act 1989. We have cited just the 1989 version, as that is what is cited by included studies. Thank you for noting the narrower range of its application (to England and Wales), which we have now specified.

---

## [Editor Report · Decision Letter 2]

26 Sep 2022

Foster children’s perspectives on participation in child welfare processes: A meta-synthesis of qualitative studies

PONE-D-21-12708R2

Dear Dr. McTavish,

We’re pleased to inform you that your manuscript has been judged scientifically suitable for publication and will be formally accepted for publication once it meets all outstanding technical requirements.

Your sincerely,

Yann Benetreau, PhD

Division Editor, PLOS ONE
---

## [Editor Report · Acceptance letter]

29 Sep 2022

PONE-D-21-12708R2 

Foster children’s perspectives on participation in child welfare processes: A meta-synthesis of qualitative studies 

Dear Dr. McTavish:

I'm pleased to inform you that your manuscript has been deemed suitable for publication in PLOS ONE. Congratulations! Your manuscript is now with our production department. 

Kind regards, 

on behalf of

Dr. Yann Benetreau 

Staff Editor

PLOS ONE